# Immuno-Colorimetric Neutralization Test: A Surrogate for Widely Used Plaque Reduction Neutralization Tests in Public Health Virology

**DOI:** 10.3390/v15040939

**Published:** 2023-04-10

**Authors:** Sunil R. Vaidya

**Affiliations:** Virus Registry and Virus Repository, ICMR-National Institute of Virology, 20-A Dr. Ambedkar Road, Pune 411001, India; suravaidya@gmail.com; Tel.: +91-(0)20-26127301; Fax: +91-(0)20-26122669

**Keywords:** emerging and reemerging viruses, neutralizing antibody, plaque reduction neutralization test, vaccine-preventable viral diseases

## Abstract

Since their first documentation in 1952, plaque reduction neutralization tests (PRNTs) have become the choice of test for the measurement of neutralizing antibodies against a particular virus. However, PRNTs can be performed only against viruses that cause cytopathic effects (CPE). PRNTs also require skilled personnel and can be time-consuming depending on the time required for the virus to cause CPE. Hence, their application limits large-scale studies or epidemiological and laboratory investigations. Since 1978, many surrogate PRNTs or immunocolorimetric assay (ICA)-based focus reduction neutralization tests (FRNT) have been developed. In this article, ICAs and their utility in FRNTs for the characterization of neutralizing antibodies, homologous or heterologous cross-neutralization, and laboratory diagnosis of viruses of public health importance have been discussed. Additionally, possible advancements and automations have been described that may help in the development and validation of novel surrogate tests for emerging viruses.

## 1. Developments in Detection of Viral Plaques

Renato Dulbecco was the first to demonstrate the presence of ‘virus plaques’ on a monolayer of chicken embryo fibroblasts, specifically for Western Equine Encephalomyelitis and Newcastle Disease viruses [1]. These findings were followed by reports of plaque formation and isolation of pure lines for polioviruses [2]. Subsequently, newer developments were reported for plaque detection and its successful application in plaque reduction neutralization tests (PRNTs) [3,4,5,6,7,8,9,10,11]. Tomori et al. described a plaque assay and PRNT for the highly infectious and fatal Lassa virus and its utility in passive immunization for the treatment of Lassa fever [12]. For many decades, PRNTs have been considered ‘a gold standard test’ for the characterization of functional neutralizing antibodies (Nt-Abs) to specific viral agents [13]. Neutralization tests (NTs) utilize live viruses to assess virus-serum interactions, indirectly measuring the level of Nt-Abs [10]. NTs are mainly established for viruses that produce distinct cell cytopathic effects (CPE), for partially grown viruses in different cell lines, and for a few viruses that do not show any evident CPE. In the PRNT, the cell monolayer is stained with various stains, i.e., crystal violet, neutral red, methylene blue, amido black, and naphthalene black, and the clear zone (plaque) formed due to cell CPE is counted, recorded, and the Nt-Abs titers are deduced using a statistical formula [10]. Ultimately, the quantity of live-challenge virus neutralized by the diluted serum (Nt-Abs) is measured. Two types of PRNTs have been described, wherein the first method test serum dilutions are varied and challenged with a fixed amount of virus, and in the second method, a test serum dilution is fixed, and the amount of challenge virus is varied.

For the first time, Okuno’s group has described peroxidase-anti-peroxidase (PAP) staining-based NTs, and afterwards, immunostaining-based methods on various cell substrates have been standardized for other viruses of public health importance [14,15,16,17,18]. A human epithelial cell (Hep-2C)-based sensitive, specific, and rapid method was described for detection and identification of polioviruses in a large number of clinical specimens [19]. The majority of immunostaining-based methods [Table 1] require a specific cell fixative (formaldehyde, paraformaldehyde, glutaraldehyde, methanol, methanol-acetone, ethanol, acetone, etc.), a cell permeabilization reagent (Triton-X 100, NP-40 solution, Ethanol, Methanol, etc.), a blocking buffer (containing bovine serum albumin, skimmed milk powder, fetal bovine serum, Tween-80, etc.), a primary antibody (monoclonal or polyclonal), an alkaline phosphatase or peroxidase-conjugated secondary antibody, and a particular substrate (3,3′, 5,5′-Tetramethylbenzidine, Nitro blue Tetrazolium plus X Gal, 3,3′-Diaminobenzedene tetrachloride, 3-amino 9-ethylcarbazole, etc.) for the immuno-precipitation of viral foci. All of the published immuno-staining-based assays described in Table 1 were based on various cell substrates (i.e., epithelial, endothelial, fibroblast, and neuroblastoma), These assays also required various incubation periods for virus-serum interactions, primary or secondary antibody binding, and the assay completion time may vary between 12 h to 13 days after infection, depending on the appearance of visible and countable virus foci. Afterwards, for the detection of viruses directly from the clinical samples or cell culture fluids (virus isolates), many investigators utilized immunocolorimetric assay (ICA)-based methods. Detection of virus-infected cells and their quantifications were documented for the wild-type polioviruses and rubella viruses using cell passaged soup and clinical specimens [19,20]. These studies describe the utility of ICAs for their respective surveillance programs and pre- and post-vaccination studies. Subsequently, ICA has been employed for the detection of measles, mumps, and rubella viruses in the clinical specimens collected from suspected cases and cell culture-grown viral isolates [21,22].

## 2. Plaque- and Focus-Based Neutralization Tests

A PRNT measures the level of virus-specific Nt-Abs in a serum or plasma sample by determining the ability of various dilutions to block the production of viral plaques on cell monolayers by a known amount of live virus [7]. A PRNT is a more sensitive quantitative assay than other serological methods, and is preferred for the accurate measurement of immune responses during vaccine trials or epidemiological studies. PRNTs have proven to be a useful tool for detecting low levels of Nt-Abs due to their high sensitivity over HI and ELISA-based serological assays that may fail to detect weak antibody responses in infants or younger children due to their immunological immaturity [10]. Despite their superior sensitivity, PRNTs have some disadvantages. They are time-consuming to perform and have incubation periods of 5–7 days or more. Therefore, testing large numbers of sera samples is labor-intensive and requires skilled operators. Like any biological assay, a PRNT is inherently variable and cumbersome to standardize, and it can be operator-dependent [10,21]. Both focus- and plaque-based methods work on the same principle and differ only in the surface area of the tissue culture plates used, the volumes of reagents added, and in the final staining and visualization steps (Figure 1). A Focus Reduction Neutralization Test (FRNT) is performed in a 96-well format; therefore, it enables the use of multichannel pipettes for dispensing cells, preparing serum dilutions, and transferring reaction mixes in micro-titer plates. This method results in greater inter- and intra-assay reproducibility than in a PRNT. The micro-titer well format allows a greater number of samples to be tested within a plate in a FRNT compared to a PRNT. FRNTs also consume less sample volume, reagents, and take up less space in CO_2_ incubators, reducing the overall assay cost and saving time with an increased output. The precipitation of viral foci on cell substrates by primary and secondary antibodies used in FRNTs results in a faster NT than plaque staining using crystal violet or other stains. Though both FRNTs and PRNTs showed similar sensitivity with inter-assay variation (Nt-Ab titers) of two- and three-fold, respectively, FRNTs offer the advantage of speed, reduced sample volume, and the possibility of automation using 96-well plates [21,22]. Other studies have also indicated good correlations between PRNT- and FRNT-derived Nt-Ab titers for Japanese encephalitis viruses (JEV) [29]. Thus, FRNTs offer an added advantage for studying JEV vaccine efficacy and subsequently for other Vaccine Preventable Diseases (VPDs).

## 3. Measurement of Neutralizing Antibodies to Public Health-Important Viruses

Generally, for the characterization of the qualitative and quantitative immune responses against wild-type and vaccine strains, in-vitro NTs are the preferred methods. For various viruses, viral plaque or focus reduction-based NTs have been standardized, validated with established virus-specific IgG Enzyme Immuno-Assays (EIAs), and employed to measure the Nt-Abs response. NTs have been utilized for confirmation of recent or past infections (with known or unknown etiology). In addition, modified NTs based on the principle of fluorescence detection and pseudo-viruses or recombinant-viruses have also been explored. The level of Nt-Ab titers that protect against the acquisition of viral infection and development of clinical disease have been documented using a sensitive PRNT for mumps and measles viruses [6,65]. Throughout the COVID-19 pandemic, laboratories in the USA have initiated collaborative networks to develop, validate, and improve serological methods and related technologies, and to share these advances with different institutions. Such comparative studies have demonstrated variability the in results of live-virus neutralization, pseudo-virus neutralization, and surrogate NTs. Hence, the characterization, comparison, and harmonization of serological assays and their performance is crucial in virology laboratories [66].

There are methods in place to characterize functional Nt-Ab responses to the Poliovirus and Yellow fever virus, for which well-tested and proven vaccines are available [2,11,19,49,67]. Different types of NTs based on either viral plaque or focus staining procedures have been developed for the VPDs, i.e., Measles virus (MeV), Mumps virus (MuV), Rubella (RuV) and Varicella Zoster virus [7,10,13,16,21,22,38,39,40,41,42,43,44,45,55,56,65,68]. Similarly, many improved versions of NTs have been described for JEV, especially for vaccine and wild-type strains [14,15,27,28,29]. The applications of various NTs have been documented for Rotaviruses that can be easily prevented by immunization [69,70]. Interestingly, rapid and reliable NTs are crucial to understanding the homologous and heterologous immune response in the vaccine recipients and are crucial to understanding the vaccine’s effectiveness and any antibody decay. In developed countries, vaccines for influenza viruses are available through routine immunizations; hence, rapid FRNTs and their usefulness for nasal wash specimens are crucial [18]. An automated FRNT was described for the characterization of immune responses to influenza viruses amongst immunized subjects [46]. For the Respiratory Syncytial virus (RSV) and the Crimean-Congo Hemorrhagic Fever virus (CCHFV), ICA-based NT and pseudo-PRNT has been standardized [23,62]. Interestingly, vaccines are under development for RSV, and limited use vaccines are available for CCHFV. Likewise, NTs have been developed for the measurement of immune responses to Severe Acute Respiratory Syndrome Corona virus-2 (SARS-CoV-2) variants [33,35,66].

Similarly, various types of NTs, i.e., FRNTs, micro-NTs, fluorescent-FRNTs, immunostaining based PRNTs, PAP-NTs, and automated versions of FRNTs, have been established by various investigators for Hantavirus, Bourbon virus, West Nile Virus (WNV), Zika virus (ZIKV), Dengue virus (DENV), and Chikungunya virus (CHIKV), either for vaccine developments, epidemiological investigations, sero-prevalence studies, or laboratory diagnoses [8,14,17,25,26,27,30,31,32,48,51,52,53,54,63,71,72]. These viruses cause infections in the form of sporadic cases, outbreaks, or epidemics which lead to significant morbidity and mortality. Likewise, improved NTs have been described for other viruses of public health importance, i.e., Metapneumovirus, Cytomegalovirus, Adenovirus type 7, Coxsackievirus B3, BK virus, and Hepatitis C virus [24,57,58,59,64,73]. In addition, the application of NTs for Infectious anemia and Herpesvirus type-1 animal viruses has been described [60,61], indicating their utility for a one-health program. Recently, many countries are working on a ‘one health approach’ to study various infectious/zoonotic diseases.

## 4. Sero-Epidemiological Investigations, Vaccine Trials, and Cross-Neutralization Studies

Previously, many sero-epidemiological investigations of viruses of public health importance, i.e., DENV, CCHFV, Rift Valley fever virus, WNV, Usutu virus, CHIKV, Bourbon virus, Hantavirus, ZIKV, and SARS-CoV-2, have been carried out to confirm the presence of virus-specific IgM and IgG antibodies, and subsequently, case confirmation using gold standard PRNTs or by virus isolation [74,75,76,77,78,79,80,81,82,83,84]. Due to cross-reactivity amongst the arboviruses, cell culture-based NTs are the preferred serological tool for case confirmation. PRNT is a widely utilized serological reference test for assessing the performance of newly developed IgM and IgG ELISAs [83], MMRV multiplexed immunoassays [85], SARS-CoV-2 lateral flow immunoassays [86], and for assessing Rotavirus vaccine performance [87,88,89]. PRNT has also been widely used for antigenic characterization and cross-neutralization studies of poliovirus serotypes [11], Coxsackie B strains [90] and hantavirus serotypes [91]. Despite the accessibility of commercial ELISAs, the NTs are being utilized in sero-epidemiological investigations for different viruses.

The comparative dengue seroprevalence in children (*n* = 2996) from Philippines was studied using IgG ELISAs and FRNTs. Although high sensitivity and specificity (>90%) were documented for ELISAs, their false-negative and false-positive results limit their use for pre-vaccination screening [26]. This could be due to the type of antigen (i.e., whole/disrupted virus particles, purified target protein, or recombinant protein) utilized in the coated ELISA plates. The prevalence of CHIKV infection in suspected dengue patients was studied amongst six countries using FRNT, and the results were compared with IgM and IgG ELISAs [32]. For the confirmation of acute hantavirus infection in suspected leptospirosis cases from the Netherlands, use of FRNTs was documented successfully [54]. A subset of serum samples (*n* = 53) referred for MeV diagnosis were tested using FRNTs, of which 47.1% of cases showed a correlation with measles IgM antibody detection [40]. Similarly, the utility of FRNTs was studied for the laboratory diagnosis of MuV in parotitis patients (*n* = 80), where 58.8% of serum samples showed positivity in all three tests, i.e., IgM ELISA, IgG ELISA and FRNT [42]. Such studies would help to understand the appearance of different classes of antibodies in the symptomatic and asymptomatic cases coupled with rapid and sensitive molecular methods. The WNV-specific Nt-Abs were detected by fluorescent-FRNT in 21 of 145 wild bird serum samples, indicating WNV prevalence in the Eastern region of Russia [30]. Such studies are crucial to monitor the virus transmission pathways within and between different countries due to migratory birds.

Recently, a study investigated vaccine-induced antibody responses in patients with non-Hodgkin lymphoma, including chronic lymphocytic leukemia, against live SARS-CoV-2 (WA1/2020), B.1.167.2 (Delta), and B.1.1.529 (Omicron) strains [34]. Interestingly, FRNT titers were found to be comparatively lower in the patients than in immunized healthy individuals. A FRNT-based cross-neutralization study indicated that the hybrid immunity developed by vaccination (Pfizer’s BNT162b2 or Johnson & Johnson’s Ad26.CoV2.S) and Omicron BA.1 infection offers protection against Delta and other variants of SARS-CoV-2, whereas immunity developed due to infection with Omicron BA.1 alone offers limited cross-protection [36]. A significant reduction in the FRNT titers of both vaccine-induced and vaccine-plus-infection was observed for the Omicron variant [35]. In the context of SARS-CoV-2 infections or reinfections due to different strains, such surrogate NTs play an important role in understanding the decay of vaccine-induced immunity (after one dose, two doses, or booster doses with different types of vaccines) to measure the functional Nt-Abs.

A study from Iran measured homologous and heterologous Nt-Abs to MeV A, B3, D4, and H1 genotypes among measles-immunized children, indicating sufficient levels of Nt-Abs titers against the circulating genotypes [68]. Such studies play a useful role in the context of the global measles elimination goal (and goals for elimination of other diseases) to determine the waning of vaccine-induced immunity (if any), or to determine antigenic differences between vaccine and circulating wild-type viruses. Applications of FRNTs for MeV, MuV, and RuV cross-neutralization studies between wild types and vaccine strains were studied in India, which are ultimately crucial for understanding qualitative and quantitative immune responses to VPDs [41,43,44]. A report from the Netherlands detected cross-Nt-Abs in measles, mumps, and rubella (MMR) vaccine recipients by utilizing MuV genotypes A, D, and G-based FRNTs [39]. Gouma et al. showed significantly lower FRNT titers to MuV genotype A (Jeryl Lynn strain) in pre-outbreak serum samples (*n* = 50) obtained from symptomatic, asymptomatic, and non-infected subjects, suggesting the possibility of a strain-specific neutralization pattern. Similarly, cross-neutralization studies using MuV genotypes A, F, H, and I challenge viruses have been carried out from Korea, which revealed that post-vaccination FRNT titers in the vaccine recipients (MMR one dose) were significantly lower against wild types compared to vaccine strains [38]. Such studies are crucial because the majority of the countries (where mumps-containing vaccines are available through their national immunization programs) are experiencing outbreaks due to MuV genotype G viruses, and current vaccines (the majority) are based on the genotype A strains. This also applies to the other viruses of public health importance for which vaccines are available.

## 5. Recent Developments and Automations

Recently, various types of liquid handling systems have become available in the market that can be utilized for NTs. In addition, viral plaques or foci stained with various dyes or fluorescent reagents can be captured using high-resolution digital cameras, and results can be recorded using advanced instrumentation systems. In-built statistical formulas in the software can be used to obtain Nt-Ab titers, enabling results to be recorded in a digital format. An automated colorimetric micro-NT was developed for WNV which utilizes the principle of neutral red dye retention and records results on a spectrophotometer using advanced software [71]. An improved and rapid version of the NT was standardized for RSV using an automated plaque counting system, which showed a high correlation (R^2^ = 0.95) between manual and automated methods with Nt-Ab titer differences within two-fold [23]. A similar semi-automated ICA-based NT has been described for equine infectious anemia virus [60]. A rapid and efficient NT based on ELISA was standardized for Coxsackievirus B3 that showed good correlation (R^2^ = 0.94) with the CPE-based NT [24]. Similarly, blue-stained viral foci of herpes simplex virus type-1 on Human U-2 OS cells were captured using an automated analyzer that eventually resulted in a reduced assay time [50]. A study by Lin et al., demonstrated the utility of an improved micro-NT using a flatbed scanner and a well-plate reader software for measurement of Nt-Ab response against Influenza A(H1N1)pdm09, A(H3N2), and B viruses [47]. A replication-competent luciferase-secreting DENV reporter was generated by Saipin et al., and afterwards an NT was standardized and compared with FRNT on a panel of serum samples that revealed a good agreement [92]. Recently, the Viridot program showed the process for standardization of manual and Viridot plaque counting methods, and its performance was evaluated on a variety of plaque images which revealed comparable Nt-Ab titer outputs [72]. Recently, two advanced methods for evaluating the neutralization capacity of SARS-CoV-2 antibodies were described, of which one was based on immunostaining of infected cells with chromogen (FRNT), and the other was based on an infectious clone-derived SARS-CoV-2 virus expressing a fluorescent reporter (mNeonGreen based FRNT) [33]. Both protocols are useful in a high-throughput setting and large-scale vaccine studies or clinical testing. Thus, the advancements and automation of FRNTs are possible and useful for public health virology. Furthermore, the cost of FRNTs can be reduced through automation, ultimately improving the quality/quantity of scientific data.

## 6. Important Unresolved Issues and Way Forward

A large number of commercial companies are producing monoclonal antibodies against various viruses using different approaches; thus, the cost of such antibodies may be lowered, and ultimately, the assay expenses could be reduced. In addition, the use of secondary IgG antibodies may be excluded by conjugating virus-specific monoclonal antibodies directly with different peroxidases or other enzymes, which will ultimately help in reducing the assay time. Moreover, there is a great need for the development of ‘universal’ cell-fixative and permeabilization reagents, blocking buffer solutions, and very specific primary/secondary antibodies that can efficiently bind to target virus surface proteins. However, optimization of various steps (i.e., virus-serum incubation, cell-fixation, cell-permeabilization, blocking of cell monolayer, application of various antibodies/concentrations, and use of a particular substrate for visualization) is critical during standardization of immuno-staining protocols and ICA-based NTs. Studies on bovine herpesvirus 1, indicated virus-serum incubation (i.e., incubation period) as crucial step for accurate measurement of antibody levels [93,94]. Thus, optimization of all the steps in NTs is essential before they are included in particular studies. Large-scale sero-epidemiological studies based on gold standard NTs can be performed by utilizing automation (automated liquid handling system, plate washers, and image analyzers). Additionally, mobile BSL-2/3 laboratories with basic cell culture facilities can be equipped in countries where the population resides in difficult terrains. Such an integrated approach may be explored under the ‘One Health program’ so that various infectious diseases can be studied, and effective control measures can be implemented [95]. For the differential diagnosis of various viral infections with inconclusive clinical presentations, and for large-scale sero-epidemiological/vaccine studies, ICA-based rapid and reliable NTs can be explored.

## Figures and Tables

**Figure 1 viruses-15-00939-f001:**
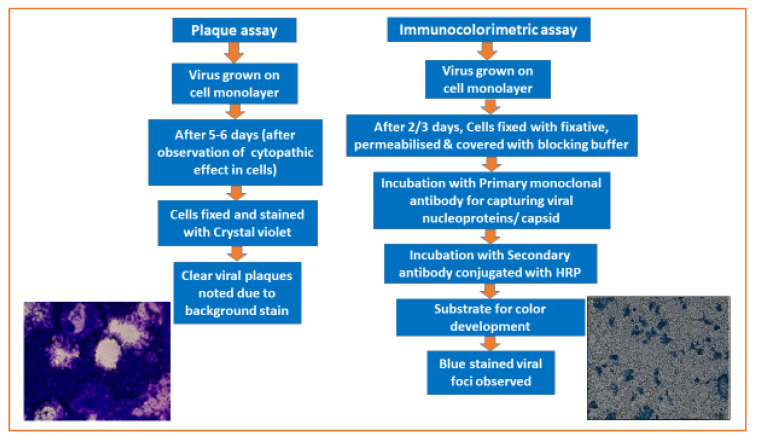
Detection of the mumps virus grown on Vero cell substrates using a plaque assay (after 5 days) and an immunocolorimetric assay (after 2 days).

**Table 1 viruses-15-00939-t001:** Published immuno-staining-based neutralization tests for various viruses important for public health.

Virus/Strain	Virus Detection Automation/Manual	FRNT or Similar Modified Protocol	Cell Substrate	Assay Time	Samples Tested EPD/Vaccine	Reference
Respiratory Syncytial virus (A2 strain)	Immunospot analyzer	FRNT	Vero CCL-34	3 days	Pooled serum panel	[23]
Coxsackievirus B3(XM08-2035)	Immunospot analyzer	Nt-ELISPOT/ FRNT	RD	12 h	Suspected HFMD cases	[24]
Dengue Virus (DENV)	Imaging cytometer	FRNT	Vero CCL-81	1 day	Sera of 81 DEN-4 cases	[25]
DENV (serotype 1, 2, 3, 4)	Immunospot analyzer	FRNT	Vero-81	48 h	Longitudinal seroprevalence	[26]
Dengue (1–4) Japanese Encephalitis Virus (JEV) (Nakayama, Beijing)	Manual	Micro FRNT	C6/36	Not Available	Dengue negative & vaccinated 102-sera	[27]
JEV	Manual	FRNT	Vero Osaka	46 h	133 JEV PRNT positive sera	[28]
JEV (Nakayama, Beijing vaccine)	Immunospot analyzer	FRNT	Vero-E6	48 h	39-sera (Unvaccinated) 53-sera (JE vaccinated)	[29]
JEV	Manual/Microscope	FRNT	BHK-21	24–28 h	20 sera (healthy adults)	[15]
West Nile (NY99-6922)JE (Genotype-1)	Microscope/ Manual	FRNT	BHK-21 CCL-10	24 h	145-wild birds sera	[30]
West Nile virus	Manual	FRNT	Vero CCL-81	2-days	Not Available	[31]
Chikungunya virus (S-27 Africa, RRV T48)	Stereomicroscope	FRNT	Vero	30 h	800-sera from suspected dengue cases	[32]
Severe Acute Respiratory Syndrome Coronavirus-2 (SARS-CoV-2)	Immunospot analyzer	FRNT	Vero-E6	3 days	Plasma samples 9-healthy controls 10-convalescent	[33,34]
SARS-CoV-2 (ancestral B.1, delta, omicron)	AID iSPOT reader	FRNT	Vero-E6	24/ 32 h	80-sera (immunized with different vaccines)	[35]
SARS-CoV-2 (ancestral B.1, delta, omicron)	Immunospot analyzer	FRNT	Vero-E6 H1299-E3	18 h	39-sera (vaccinated with different vaccines)	[36]
SARS-CoV-2	Immunospot analyzer	FRNT	Vero WHO	24 h	Methodology (basic study)	[37]
Mumps virus (MuV)	Immunospot analyzer	FRNT	Vero hSLAM	48 h	50 MMR (1 dose) Vaccinated sera	[38]
MuV	Manual	FRNT	Vero	40 h	105-serum samples (pre/post-outbreak) with two doses of MMR	[39]
MuV (Enders, Miyake, Urabe)	Manual	FRNT	Vero	2 days	17-sera (lab confirmed mumps)	[16]
Measles virus (MeV) MuV Rubella virus (RuV)	Manual	ICA/ FRNT	Vero hSLAM Vero Vero hSLAM	2 days 2 days 3 days	Serum/ throat swabs from suspected MMR cases	[21,22,40,41,42,43,44]
MeV	EliSopt Reader Image Analyzer	FRNT	Vero (ECCC UK)	28 h	90-serum samples 38 IV-Ig-products	[45]
Influenza virus (5 Strains) NYMC X-179A (H1N1) A/California/7/2/2009)	EliSopt Reader Image Analyzer	FRNTELISA-TCID_50_	MDCK	18–20 h	108 serum samples (Immunized subjects, Laboratory workers & Clinically suspected)	[46]
Influenza A and B virus	Microscope	FRNT	MDCK	24 h	60-serum samples	[18]
Influenza A and B viruses	Flatbed scanner	FRNT	MDCK MDCK-SIAT	Overnight	Methodology	[47]
Zika virus	ELISPOT Reader	FRNT	Vero WHO C6/36	40 h	Serum panel	[48]
Yellow fever virus (YF-17-D)	ELISPOT Reader	FRNT	BHK-21	4 days	15-sera (vaccinated healthy volunteers)	[49]
Poliovirus	Manual	CPE Blue-Cell ELISA/ICA	Hep-2C RD L20B, Vero	24 h 48 h 7 days	97 poliovirus isolates 43-isolation positive specimens	[19]
Herpes Simplex Virus 1 (KOS strain)	Immunospot analyzer	ELISOPT-NT PRNT	U-2 OS	14 h 3 days	269-Sera (healthy individuals)	[50]
Hantavirus (7 strains)	Manual	FRNT	Vero-E6	9–12 days	190-sera 17-Mammalogists sera Seroepidemiology (Lativa, *n* = 333)	[51,52,53]
Hantavirus (DOBV Slovenia, SEOV 80-39, PUUV Kazaan)	Manual	FRNT	Vero-E6	7–13 days	22-serum samples	[54]
Varicella-zoster (VR 841)	Manual	Immuno Peroxidase technique based PRNT	WI-38	72 h	8-serum samples	[55]
Varicella zoster virus	Immunospot analyzer	FRNT	ARPE-19	3-days	Mouse serum (53-immunized & 16-non-immunized)	[56]
Hepatitis C virus (JFH-1 HCV 2a)	Manual	FRNT	Huh-7	3-days	77-sera (57-chronic HCV patients)	[57]
Human polyomavirus BK (BKV, prototype Gardner)	Microscope/ Manual	Immunoperoxidase-NT	Vero	6-days	64-serum samples	[58]
Human Metapneumovirus (CAN 97-83, group-A & CAN 98-75, group-B)	Manual	FRNT	LLC-MK2	5-days	20-serum samples	[59]
Equine Infectious Anemia virus (PV & D9 strains)	Immunospot analyzer	FRNT	Fetal Equine Kidney	72–96 h	Serum panel (*n* = 6)	[60]
Equine Herpesvirus type-1 (EHV-1 strain 89C25p)	Manual	FRNT	MDBK	24–48 h	Foal serum (*n* = 30) Infected Horse sera (*n* = 16)	[61]
Crimean-Congo hemorrhagic fever virus (Turkey-Kelkit06)	Microscope/ Manual	FRNT (both ICA/IFA)	Vero-E6	3-days	69-serum samples (20-acute & 49-convalescent)	[62]
Bourbon virus	Immunospot analyzer	FRNT (both ICA/IFA)	Vero-E6 CCL81	24 h	440-serum samples	[63]
Human cytomegalovirus (HCMV AD169)	Immunospot analyzer	FRNT	MRC-5 HEL-299 WI-38	14–20 h	Serum panel (*n* = 57)	[64]

## Data Availability

Not applicable.

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
