# Peer review of "Immuno-Colorimetric Neutralization Test: A Surrogate for Widely Used Plaque Reduction Neutralization Tests in Public Health Virology"

_viruses, 2023, doi:10.3390/v15040939_

Round 1
Reviewer 1 Report
The concept of the review by Sunil Viadya is good due to the increased interest into the utility of using PRNT, in particular how the PRNT data for the efficacy of SARS CoV 2 vaccines against the emerging variants became in the public domain. Whilst the review is comprehensive I have a few suggestions and queries.
· I would include a brief description of both PRNT and FRNT, potentially incorporating figures which would set the scene for those unfamiliar with both methodologies
· There were many occasions and linked references when the author stated that PRNT/FRNT was used to conform the presence of IgG and IgM response. However, as both PRNT and FRNT are total antibody assays they confirm the presence of neutralizing antibody but cannot conform the class of antibody.
· I would agree that the PRNT/FRNT are used to confirm the presence of antibody detected in EIAs. However, I would argue that the greatest benefit of PRNT/FRNT is the ability to investigate the specific neutralizing antibody response to different genotypes or emerging subtypes which is very difficult when using EIAs. This was covered in page 6, lines 161-182 but I feel needs expanding. In addition the importance of PRNT/FRNT for investigating vaccine response should be dealt with in more detail
· On closer examination of the reference, especially page 5, I am not convinced that your references the author quoted actually support the statement in the text, such as page 6 lines 111-118
· There are occasions when the text is not supported by any references Page 4 line 63 to 68
Author Response
Attachment please.

Reviewer 2 Report
The focus of the review indeed was on the neutralizing antibodies in public health rather than the methods. The author did not think through the reasoning when comparing the assays/methods (for example, the nature of viruses and cell growth) and did not make fair justifications (for example, the cost effectiveness (reagents and instrumentation) and labor intensiveness of ICA). In addition, ICA-based methods can be applied to virus surveillance other than measuring neutralizing antibodies. The author should look for references more carefully beyond neutralization test when thinking about uses in public health.
Other comments:
1. Line 63-66: Author said “PRNT is more sensitive than other serological methods as it is quantitative assay and preferred for the accurate measurement of immune response during vaccine trials or epidemiological studies. PRNT has proven to be a useful tool for detecting low levels of Nt-Abs due to its high sensitivity over serological assays” It is not clear what is being compared: what are the other serological methods that measure inhibition of virus growth that the author referred to? If it meant IgG ELISA, the author should avoid comparing assays of different natures directly.
2. Line 66-67: Author mentioned “weak antibody response” and “immunological immaturity”. It is not clear what the author tried to address. Please be more specific and provide good and related references.
3. Line 71-72: Comments on PRNT “inherently variable, cumbersome, operator dependent” (Line 71-72), and FRNT “a faster NT than plaque staining using crystal violet…” are very subjective.
4. Line 139: Reference 80 did not compare to neutralizing titer; Line 141: the study did not compare to FRNT.
5. Lines 179-182: the sentences indicate that the section is not completed.
Author Response
Attachment please.

Reviewer 3 Report
Vaidya, S.R. wrote a review entitled "Immuno-colorimetric Neutralization Test: A surrogate for widely used Plaque Reduction Neutralization Test in Public Health Virology", in which he describes the technique of "Plaque assay-based plate reduction neutralization tests (PRNT)" and the surrogate "Focus reduction format neutralization tests (FRNT)" and compare test results against different viruses. There is even a descriptive table with the main FRNT assays, in which viruses were applied, the type of cell used, incubation time, type of sample used, and reference. The article explores the advantages of using a neutralization test, which evaluates neutralizing antibodies, compared to ELISA tests and other methods. The use of PRNT in the analysis of vaccine efficacy is also discussed, something very important, in which this technique stands out as the gold standard. Finally, the author comments on recent advances involving colorimetric markings to increase sensitivity and analysis time and comments on the limitations involving the lack of reagents or structure (BSL-2/3 laboratories) to perform the test. Overall, the article is well-written and the topic is important. In order to collaborate with the study, there are two important issues that must be addressed in this review when it comes to a test to assess the presence of antibodies. The first is in relation to the virus-serum incubation time, although the author has added the test run time in Table 1, he does not discuss this difference. It is known that the incubation time influences the sensitivity of the test, and is therefore a factor to be analyzed and discussed in the article. Another very important point is how long the PRNT or FRNT can detect neutralizing antibodies after vaccination or natural infection. It is known that the best tests can detect antibodies in serum as early as 7 days postinfection. For more details on these two points, the author can read and cite the works of Kramps et al., 1994 and 1996 described below. Additionally, the World Organization for Animal Health also recommends a long incubation time for some tests in order to increase their sensitivity. For example: "VN in cell culture can be performed in several ways, which vary according to the length of incubation of the virus/serum mixtures (e.g. 1 hour at 37°C or 24 hours at 4°C) and the presence or absence of complement. Most laboratories use a reaction period of 1 hour at 37°C in the absence of complement, because this is easy and rapid. However, the sensitivity can be improved by increasing the incubation period to 24 hours at 4°C, which facilitates the detection of antibody levels 10–15 times lower than in the 1-hour method. For international trade purposes, the test method should be validated as being sensitive enough to detect the OIE Standard Reference Serum diluted 1/2."
Details at https://www.woah.org/fileadmin/Home/fr/Health_standards/tahm/3.01.02_AUJESZKYS.pdf.
Kramps JA, Magdalena J, Quak J, Weerdmeester K, Kaashoek MJ, Maris-Veldhuis MA, Rijsewijk FA, Keil G, van Oirschot JT. A simple, specific, and highly sensitive blocking enzyme-linked immunosorbent assay for detection of antibodies to bovine herpesvirus 1. J Clin Microbiol. 1994 Sep;32(9):2175-81. doi: 10.1128/jcm.32.9.2175-2181.
Kramps JA, Perrin B, Edwards S, van Oirschot JT. A European inter-laboratory trial to evaluate the reliability of serological diagnosis of bovine herpesvirus 1 infections. Vet Microbiol. 1996 Nov;53(1-2):153-61. doi: 10.1016/s0378-1135(96)01243-6.
Author Response
Attachment please.

Round 2
Reviewer 2 Report
Improvement is noticeable but the paper still lacks good focus. Although the author indicated that the central idea of the review is on the uses of neutralization tests (NTs) in public health, the manuscript still needs to be tidied up and reduce content that is less relevant, such as methodology and technical details (Lines 91-100 and Line 266-271). The author should look at what NTs can aid on the existing, commonly used serological methods (IgM, IgG ELISA, Multiplex Bead Assay) in surveillance/case classification, with a focus on one or two pathogens in depth and be able to comment on the challenges that the author observed from literature (for example, a cutoff for immunity and protection).
The author tended to impress readers by listing the studies but not making comments, which made the reading difficult to follow. An example is the paragraph in lines 174-192 where 6 studies comparing ELISA and FRNT were listed but there was no clear conclusion on whether NTs added any value in case classification. The following paragraph, on the other hand, was much easier to comprehend.
The manuscript is not well organized, and similarities were found in sections 3, 4, and 5. The manuscript lacks consistency. Sometimes FRNT is referred to immunostaining-based NTs but in other places, assays such as ICA-based FRNT (Line 142) or ICA-based NTs (Line 283) were mentioned.
The author needs to explain acronyms in Table 1 (mFRNT, IP-FRNT) and different NTs in line 147-148. Several names have been created for the ICA-based assays in literature although they all were essentially identical (for example Blue-cell ELISA vs ICA and CPE-based NTs vs PRNT (Line 241)). Some literature even used FRNT to refer PRNT (i.g., ref 33). The author may take this opportunity to clarify/unify the name of the assay.
The author must be careful when using adjectives and adverbs. An example is in Line 118 where it says “…slightly modified NTs based on the principle of fluorescence detection….” Fluorescence based NTs, especially using pseudovirus or recombinant virus, is VERY different from PRNT and FRNT because those assays use genetically modified viruses. The other example is in Line 121 where the author indicates “…have been documented using a highly sensitive PRNT for mumps and measles”. It is not clear what the “highly sensitive PRNT” is. Another example is in line 258 where it says, “a cost of FRNTs can be automatically reduced due to….” The cost cannot be “automatically” reduced.
The author should be responsible to select, cite, comprehend, and interpret references precisely. For example, Ref 10 was not relevant to weak or immature immunity (line 87). Ref 20 indicated detection of “virus infected cells”, not “live virus particles” (line 71). Reference 43 should be “automated FRNT” instead of “advanced” version of FRNT (line 140).
Other comments:
Figure 1 may not be necessary. Nevertheless, the author needs to provide information for Figure 1 (which virus and what type of cell in the example) and reference where the figure taken from.
Line 156-158:
One Health (capitalized). Needs a reference.
Line 269, 273:
It should be “virus-serum incubation”, not interaction.
Author Response
Dear Editor,
Please find attached herewith a covering letter along with the pointwise clarifications to the Reviewer-2 comments/ suggestions. Thank you.
BW,
Sunil
